# Distilling Neuro-Symbolic Programs into 3D Multi-modal LLMs

**Wentao Mo** [1]   **Yang Liu** [1]

## Abstract

Current 3D spatial reasoning methods face a fundamental trade-off: neuro-symbolic 3D (NS3D) concept learners achieve interpretable reasoning through compositional programs but are constrained to closed-set concept vocabularies and simple programs; end-to-end 3D multi-modal LLMs (3D MLLMs) could handle complex natural language and open-vocabulary concepts but suffer from black-box reasoning without explicit spatial verification. We introduce APEIRIA, a neuro-symbolic 3D MLLM to bridge two paradigms by distilling symbolic reasoning patterns into MLLMs with natural language chain-of-thought. Our three-stage curriculum progressively builds reasoning capabilities: a) 3D perception alignment grounds object visual-geometric features to the LLM, b) CoT-SFT teaches query decomposition and stepwise verification from symbolic program traces, and c) CoT-RL extends reasoning patterns to open-set concepts and deeply nested instructions. By transferring reasoning patterns rather than concept-specific knowledge, APEIRIA preserves key NS3D virtues: transparent reasoning and modular interchangeability of planning and perception components. Evaluations on grounding, question answering, and captioning show that APEIRIA surpasses prior NS3D methods and matches state-of-the-art 3D MLLMs on 3D spatial reasoning datasets, unifying symbolic methods' systematic reasoning with MLLMs' flexibility. Code is available at https://github.com/oceanflowlab/APEIRIA.

## 1. Introduction

3D spatial reasoning is fundamental to embodied AI and 3D scene understanding, and two paradigms have emerged to

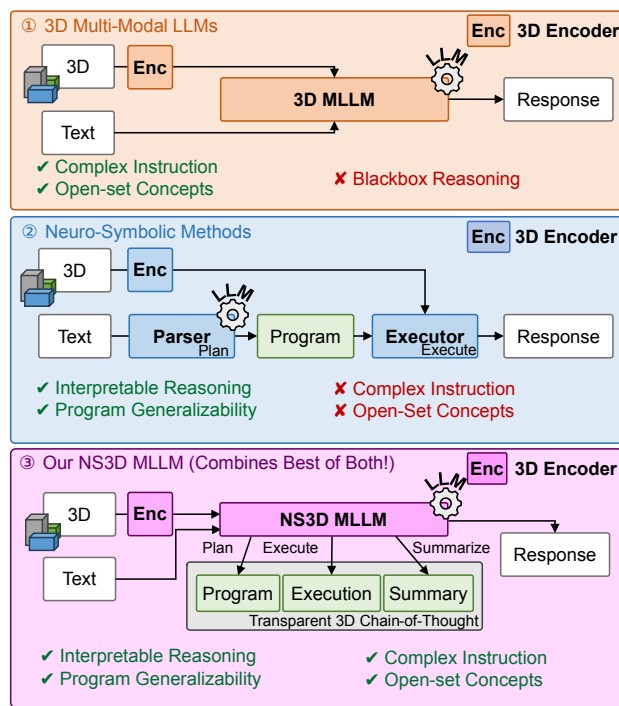

*Figure 1.* **Bridging 3D MLLMs and neuro-symbolic reasoning.** ① 3D multi-modal LLMs could handle complex natural language and open-vocabulary instructions but reason as black boxes without interpretable verification. ② Neuro-symbolic 3D (NS3D) methods offer transparent, step-by-step reasoning programs but are limited to closed-set concepts and require dense procedure supervision. ③ Our method bridges them by distilling symbolic reasoning patterns from programs into transparent chain-of-thought within 3D MLLM, combining the best of both methods.

tackle this challenge (Figure 1). The dominant paradigm, **3D Multi-modal LLMs (3D MLLMs)** (Chen et al., 2024a; Huang et al., 2024b;a; Zhu et al., 2025; Zheng et al., 2025; Yu et al., 2025; Chen et al., 2024c) (Figure 1①), leverages LLMs' semantic power to handle free-form referring expressions. Yet, lacking explicit reasoning structure, these **black-box** models directly map instructions to answers without interpretable verification steps. When they fail, there is no trace to diagnose *where* the error occurred in object recognition, relation understanding, or logical composition. In contrast, a smaller line of work on **Neuro-Symbolic 3D (NS3D) Learners** (Hsu et al., 2023; Feng et al., 2024; Yi et al., 2018) (Figure 1②) pursues interpretable spatial reasoning by decomposing queries into modular programs itera-

[1]Wangxuan Institute of Computer Technology, Peking University. Correspondence to: Yang Liu <yangliu@pku.edu.cn>.

*Proceedings of the $43^{rd}$ International Conference on Machine Learning*, Seoul, South Korea. PMLR 306, 2026. Copyright 2026 by the author(s).

tively executed by trained concept-specific networks. While excelling at systematic generalization within closed-set vocabularies, they face two fundamental barriers: (**i**) rigid concept-specific networks cannot handle **open-vocabulary** spatial queries (e.g., "cozy chair", "messy desk"); (**ii**) training modular components requires **dense step-by-step supervision** unavailable beyond simple synthetic datasets.

The field appears trapped in a trade-off: interpretable but limited NS3D learners versus powerful but opaque MLLMs. However, we identify an opportunity to decouple **concept learning** from **reasoning learning**. Neuro-symbolic programs encode precise, systematic reasoning patterns (the "syntax" of thought) which can be distilled into the MLLM, while the MLLM itself provides the open-world knowledge to understand 3D semantics. On synthetic datasets like Sr3D (Achlioptas et al., 2020) where instructions are template-generated from heuristics, the symbolic programs are guaranteed to have *complete intermediate supervision*: every primitive's input and output can be constructed and verified against ground-truth annotations, which is a supervision often unavailable in real-world datasets. This provides an ideal stepping stone for transferring reasoning capabilities into 3D MLLMs: we could first distill these verified execution traces into MLLMs, teaching the model systematic query decomposition and stepwise spatial verification skills. Then, on real-world datasets where such complete traces are unavailable, we employ reinforcement learning with outcome supervision to bootstrap and extend these learned reasoning patterns to open-vocabulary concepts and complex nested instructions.

Building on this insight, we present APEIRIA[1], a neuro-symbolic MLLM for 3D spatial reasoning, that unifies the systematic rigor of symbolic methods with the semantic flexibility of LLMs (Figure 1③). Built upon an efficient object-centric representation, APEIRIA is trained through a **Curriculum-based Reasoning Distillation** framework that progressively builds spatial reasoning capabilities. In **Stage 1 (Perception Alignment)**, we teach the model to *see*, by aligning 3D visual-geometric features with the LLM's embedding space to recognize object categories, attributes, and locations. In **Stage 2 (Symbolic Reasoning Injection)**, we teach the model to *think iteratively*, by learning from **verified reasoning traces** from symbolic programs via CoT-SFT, where each trace contains explicit plans (query decomposition) and executions (spatial verification with **object IDs and locations**), providing **correct** supervision for systematic spatial reasoning. In **Stage 3 (Open-Set and Complex Reasoning Generalization)**, we teach the model to *adapt* to real-world instructions, by using CoT-RL to extend these learned patterns to **open-vocabulary concepts** and **complex nested queries** where step-by-step supervi-

---

[1] Απειρια, *Unlimited* in Greek.

sion lacks. This yields an end-to-end model that solves complex 3D spatial reasoning tasks while retaining key neuro-symbolic virtues: **inference transparency** through explicit reasoning traces, and **modularity** of planning and perception components interchangeable as stronger LLMs or 3D perception models emerge.

Concretely, our key contributions are: 1) We introduce APEIRIA, a **neuro-symbolic 3D MLLM** that bridges symbolic reasoning and modern MLLMs. By explicitly modeling reasoning processes as transparent chain-of-thought, it enables systematic, generalizable planning for 3D spatial reasoning tasks. 2) We propose a **curriculum-based reasoning distillation** framework that progressively transfers reasoning patterns from neuro-symbolic programs into 3D MLLMs: from perception alignment, through symbolic reasoning injection, to open-set generalization via reinforcement learning. 3) We demonstrate that APEIRIA surpasses previous neuro-symbolic 3D methods and state-of-the-art 3D MLLMs on 3D spatial reasoning benchmarks, while maintaining transparent reasoning and efficient generalization without excessive tokens or auxiliary architectures.

## 2. Related Work

**Neuro-Symbolic 3D (NS3D) Grounding and Reasoning.** Neuro-symbolic methods (Yi et al., 2018; Johnson et al., 2017; Vedantam et al., 2019; Mao et al., 2019) have demonstrated exceptional data efficiency and generalization in the 2D visual reasoning. In the 3D domain, approaches (Hsu et al., 2023; Feng et al., 2024) decompose spatial queries into compositional programs executed by concept-specific neural modules, enabling zero-shot transfer to novel program structures. However, NS3D methods face two limitations: (**i**) their reliance on concept-specific networks restricts them to closed-set vocabularies: a "chair" detector can't recognize "cozy furniture"; (**ii**) training modular components requires **step-by-step supervision** only available in simple synthetic datasets like Sr3D (Achlioptas et al., 2020) with at most 2-level nested programs, preventing scaling to complex real-world instructions. Our method addresses these gaps by distilling reasoning patterns into MLLMs for open-vocabulary understanding, and is the first to introduce outcome-based RL in NS3D methods to extend NS3D methods beyond procedural supervision requirements.

**3D Multi-Modal Large Language Models (MLLMs) for Spatial Reasoning.** There has been a surge of interest in integrating LLM with 3D spatial understanding capabilities (Chen et al., 2024a; Huang et al., 2024a; Mo et al., 2024; Zhu et al., 2025; Zheng et al., 2025; Yu et al., 2025; Chen et al., 2024c; Mo et al., 2025; Li et al., 2025a; Zhou et al., 2025; Zhao et al., 2025; Zhou et al., 2026). These methods range from token-efficient object-centric approaches (<1K scene tokens) (Huang et al., 2024a; Yu et al., 2025; Chen

et al., 2024a) to video-centric methods (Zhu et al., 2025; Zheng et al., 2025) (10K–40K scene tokens), yet both treat reasoning as a **black-box mapping** without explicit verification steps. When grounding fails, there is no reasoning trace to diagnose whether the error lies in object recognition, relation understanding, or logical composition. Our work aligns with the streamlined object-centric architecture for efficiency, but uniquely injects **structured, interpretable reasoning traces** that expose the model's spatial verification process, enabling transparent debugging and modular component replacement.

**Reinforcement Learning (RL) for Visual Reasoning.** Recently, RL techniques with chain-of-thought have recently been introduced to enhance MLLM reasoning beyond supervised fine-tuning and beyond supervised fine-tuning on black-box answer generation, with notable progress in 2D vision-language understanding and generation tasks (Zhang et al., 2025; Peng et al., 2025; Qin et al., 2025; Peng et al., 2026; Lyu et al., 2025; Gong et al., 2026). In the 3D domain, concurrent works 3D-R1 (Huang et al., 2025) and Scene-R1 (Yuan et al., 2025) represent initial attempts to apply RL for 3D spatial reasoning. However, these 3D approaches have critical limitations. Scene-R1 lacks any reasoning trace enforcement or initialization, making RL exploration unstable and inefficient. 3D-R1 generates CoT via LLM prompting, which is prone to hallucination and produces traces without **explicit spatial grounding**: objects are referenced vaguely rather than by precise IDs, locations, or bounding boxes, limiting utility for spatial verification. In contrast, our approach first distills **verified, spatially-grounded traces** from symbolic programs, where each step explicitly references objects by ID and spatial configuration. This structured initialization enables RL to refine and extend reasoning patterns rather than discover them from scratch, yielding more robust and interpretable spatial reasoning.

# 3. APEIRIA: Progressive Distillation from Symbolic Programs to Interpretable 3D Reasoning

## 3.1. Object-Centric 3D Representation

To equip APEIRIA with a comprehensive understanding of the 3D environment, we first encode both the visual properties of object instances and their spatial configurations. We adopt an object-centric scene representation that enables efficient reasoning with minimal input tokens ($\approx$400 tokens vs. 10k-40k in video-based MLLMs (Zheng et al., 2025; Zhu et al., 2025)). Following prior work (Huang et al., 2024a), the input 3D scene is first segmented into object instances using Mask3D (Schult et al., 2023). For each instance, we extract 3D visual-geometric features with Uni3D (Zhou et al., 2024) and 2D appearance features with

DINOv2 (Oquab et al., 2024). Spatial information (object location and dimensions) is encoded via learnable positional embeddings. The final object representation concatenates visual and spatial features, which are interleaved with instruction tokens as LLM input. This compact representation explicitly preserves object-level structure, while maintaining facilitating architectural simplicity. Please refer to the supplementary material for detailed implementation.

## 3.2. Curriculum-based Reasoning Distillation

Our core insight is that neuro-symbolic programs encode the exact reasoning patterns we wish to preserve—systematic decomposition, explicit intermediate states, and compositional execution—making them ideal supervision signals for teaching 3D MLLMs interpretable reasoning. Therefore, we propose to *distill the reasoning patterns* from NS programs into natural language chain-of-thought into 3D MLLMs. This approach eliminates the dependency on rigid, concept-specific networks, such as the separate networks for each object category (e.g., $f_{\text{chair}}$) or relation (e.g., $f_{\text{left}}$) required by traditional NS3D learners (Hsu et al., 2023), thereby allowing us to leverage the open-vocabulary semantic flexibility of LLMs while preserving systematic generalization.

This distillation proceeds through a progressive three-stage curriculum (Figure 2), transforming the model to an active reasoner: **Stage 1: Perception Alignment (Section 3.2.1).** We first teach the model to *see* by aligning 3D visual-geometric features with LLM's textual representation space, establishing the foundational ability to recognize and locate objects. **Stage 2: Symbolic Reasoning Injection (Section 3.2.2).** We then teach the model to *think* iteratively by distilling reasoning traces from symbolic programs. Through CoT-SFT, the model learns the "syntax" of spatial reasoning: how to decompose queries and verify them step-by-step. **Stage 3: Open-Set and Complex Reasoning Generalization (Section 3.2.3).** Finally, we teach the model to *adapt* to real-world scenarios. Using CoT-RL, we bootstrap the learned reasoning patterns to handle open-vocabulary concepts and complex nested instructions where procedural supervision for each step is unavailable. This curriculum efficiently bridges the gap between limited high-quality symbolic supervision and broad, real-world reasoning capabilities.

### 3.2.1. PERCEPTION ALIGNMENT

Prior to program distillation, we align 3D object representations with the LLM's textual embedding space through standard 3D vision-language pre-training (Huang et al., 2024b; Zhu et al., 2025; Mo et al., 2025). This stage trains on mixture of object-centric perception tasks: **object recognition** (identifying categories/attributes), **localization** (predicting coordinates), and **captioning** (generating descrip-

*Figure 2.* **Progressive Curriculum-based Reasoning Distillation Framework. Left: Three-Stage Simple-to-Complex Curriculum.** ① **Perception Alignment** grounds 3D visual features to textual concepts. ② **Symbolic Reasoning Injection** distills reasoning patterns from programs via CoT-SFT, providing procedural supervision for query decomposition and execution. ③ **Open-Set and Complex Reasoning Generalization** extends these patterns to complex, open-vocabulary scenarios via CoT-RL with outcome supervision. **Right: Program-to-CoT Translation.** To generate Stage ② supervision, we parse neuro-symbolic programs into execution traces, which are then **serialized** into natural language CoT. This explicitly maps symbolic operations to transparent reasoning steps (plans and executions), enabling the MLLM to learn the "syntax" of spatial reasoning.

tions), totaling approximately 193K instruction-response pairs (please refer to Section A.2 for data details). This establishes a robust foundation for 3D scene understanding, enabling the model to ground basic object concepts before tackling complex compositional reasoning.

### 3.2.2. SYMBOLIC REASONING INJECTION

Having established basic 3D understanding capabilities, we now inject systematic reasoning patterns from neuro-symbolic programs with Chain-of-Thought Supervised Fine-Tuning (CoT-SFT). NS programs offer a 'white-box' super-vision signal: every execution step provides ground-truth inputs and outputs. This allows us to teach the MLLM *how to think* (plan decomposition)—how to decompose and execute complex queries—rather than just what to answer. In addition, this pattern-level transfer is crucial for subsequent generalization to novel concepts and structures in Stage 3 (Section 3.2.3).

**Neuro-Symbolic Programs as Supervision.** We adopt the neuro-symbolic framework from NS3D (Hsu et al., 2023), which decomposes visual reasoning instructions into compositional programs comprising primitive operations. Each primitive accepts object sets and conditions as inputs and return filtered or related object sets. Specifically, `scene ()` initializes and returns the set with all scene objects $\mathcal{O}$;

`filter($\mathcal{O}$, condition)` selects objects matching specific semantic attributes (e.g., category or color);[2] and `relate($\mathcal{O}_A$, $\mathcal{O}_B$, r)` returns objects in $\mathcal{O}_A$ with spatial relation $r$ to any object in $\mathcal{O}_B$, while `relate_triple` extends this to triplet relations. These primitives compose into programs for complex queries, e.g., `relate(filter (desk), filter(wall), left)` finds desks left of walls. Crucially, while traditional NS3D methods implement these primitives via rigid concept-specific networks with heuristic designs that hinder open-set generalization, we propose to leverage the semantic flexibility of MLLMs to execute each primitive through natural language reasoning. To achieve this, we construct a corpus of plan-then-execution CoT to teach model how to reason systematically.

**Constructing Programs with Complete Execution Supervision.** We leverage existing annotated 3D datasets to build a diverse set of programs with **complete execution traces**: every intermediate execution step have verified output from ground-truth annotations.

*Level 1 (Single-step programs):* Instructions involve only single `filter` operations, such as "find all desks" (`filter(desk)`) or "find red objects" (`filter(red)`). we utilize category and attribute annotations from Scan-

---

[2]For brevity, we denote `filter(scene(),condition)` as `filter(condition)`.

These are all objects in the scene: |object_set|. Find the vase left to the computer.

[think]

I need to find the vase left to the computer.

*// Stepwise plans.*

Let's plan my next steps: Examine all objects in the scene; Find all objects of category 'vase'; Find all objects of category 'computer'; Check which vase(s) are 'left to' the computer.

*// Execution of scene() statement.*

First, let me list the objects and names: Object 0(vase), 1(vase), 2(bottle), 3(rug), ... [others omitted]

*// Execution of filter() statement.*

Among these objects, I can see the vase(s):
Object 0: At (6.41,...), size:1.60 x...[more objects omitted]
And the computer(s):
Object 4: At (1.44,...), size:0.87 x...[more objects omitted]

*// Execution of relate() statement.*

Now, I will analyze if any vase is left to the computer:
The vase (ID 0) is left to the computer (ID 4); ...[more relations omitted]

*// Summarization.*

From analysis above, the vase (ID 0) is left to a computer (ID 4). Now, I will formulate the response based on the identified objects.

[answer]

I found the vase left to the computer: Object 0: At (1.15, 6.09, 1.33), size: 0.86 x 0.99 x 1.79

*Figure 3.* Example CoT for a `relate` program.

Net (Dai et al., 2017) and MMScan (Lyu et al., 2024) to provide ground-truth outputs for each `filter` operation, teaching the model basic visual-geometric semantics.

*Level 2 (Two-step programs):* For relational reasoning, we utilize the Sr3D dataset (Achlioptas et al., 2020), which contains synthetic instructions guaranteed to be solvable in exactly two steps. Instructions in Sr3D follow these fixed compositional patterns, and can be directly mapped to two-step programs: (**i**) "find A with relation r to B" (`relate(filter(A),filter(B),r)`); (**ii**) "find A with relation r to B and C" or "find A with relation r with B, facing C" (`relate_triple(filter(A),filter(B),filter(C),r)`). The synthetic nature of Sr3D allows us to derive complete intermediate supervision: the outputs of `filter` steps are verified by object annotations, while the final `relate` outputs are verified by target annotations. In total, we construct 78K Level-1 traces from ScanNet and MMScan annotations and 66K Level-2 traces from Sr3D, yielding 144K verified CoT examples for this stage.

However, this reliance on perfect intermediate supervision limits scalability to real-world datasets (e.g., ScanRefer), where ground-truth execution procedural supervision are **unavailable for instructions corresponding to level 2+ programs**, necessitating the adaptive reinforcement learning introduced in Stage 3 (Section 3.2.3).

**Composing Chain-of-Thought from Programs.** As il-

lustrated in Figure 2 (right), we **translate program into natural language CoT** through parsing and serialization. For each program, we parse its abstract syntax tree into an execution sequence $\mathcal{S} = \{s_1, \ldots, s_n\}$. Each step $s_i$ is serialized into natural language via two components: (**i**) a *plan* describing the reasoning goal (e.g., "Find all objects of category 'vase' "), and (**ii**) an *execution* presenting the step's inputs and outputs with object details (IDs, positions, sizes). We use predefined templates that fill in ground-truth object sets at each step. The final CoT concatenates all plans followed by all executions, creating a transparent reasoning trace from query to answer (Figure 3). Crucially, our CoT is **spatially grounded**: each object is referenced by a unique ID with explicit locations, enabling disambiguation of same-category objects and precise spatial verification. The details are provided in Algorithm 1 in the supplementary material.

**Training Objective.** We train APEIRIA's parameters $\theta$ to generate CoT traces via standard language modeling:

$$\mathcal{L}_{\text{CoT-SFT}} = -\mathbb{E}_{(q,\text{CoT},A)\sim\mathcal{D}} \left[ \log p_\theta(\text{CoT}, A \mid q, \mathcal{O}) \right] \quad (1)$$

where $q$ is the task instruction, CoT is the reasoning trace, $A$ is the final answer, and $\mathcal{O}$ represents scene object features. This objective teaches the model to internalize systematic decomposition patterns while maintaining transparency through explicit intermediate steps.

### 3.2.3. OPEN-SET AND COMPLEX REASONING GENERALIZATION

Finally, we employ Chain-of-Thought Reinforcement Learning (CoT-RL) to extend Stage 2's systematic reasoning to handle real-world scenarios (e.g., ScanRefer (Chen et al., 2020) and Multi3DRefer (Zhang et al., 2023)) involving open-vocabulary concepts (e.g., "comfortable") and deep nesting where intermediate supervision is absent. For example, "find the comfortable furniture on the kitchen counter and besides the white fridge" requires understanding open-set attributes ("comfortable") and complex spatial reasoning beyond programs with 2 nesting level. Our CoT-RL stage addresses this challenge via Group Relative Policy Optimization (GRPO) (Shao et al., 2024) with a composite reward function tailored for spatial grounding.

First, to overcome the sparsity of IoU metrics which provide no guidance for disjoint predictions, we introduce a **Soft Grounding Reward** that evaluates configuration similarity:

$$R_{\text{grounding}} = \underbrace{e^{-\alpha \|\boldsymbol{x}_{\text{pred}} - \boldsymbol{x}_{\text{gt}}\|_2}}_{\text{location similarity}} + \underbrace{e^{-\alpha \left\| \frac{\boldsymbol{s}_{\text{pred}} - \boldsymbol{s}_{\text{gt}}}{\boldsymbol{s}_{\text{gt}}} \right\|_1}}_{\text{size similarity}} \quad (2)$$

where $\boldsymbol{x}_{\text{pred}}, \boldsymbol{x}_{\text{gt}} \in \mathbb{R}^3$ are predicted and ground-truth object centers, $\boldsymbol{s}_{\text{pred}}, \boldsymbol{s}_{\text{gt}} \in \mathbb{R}^3$ are object sizes, and $\alpha = 2$ controls decay rate. This offers dense feedback based on spatial proximity, enabling effective exploration even when

initial predictions yield zero overlap. Concurrently, to prevent the policy from collapsing into direct answering and losing interpretability, we incorporate a binary **Format Reward**: $R_{\text{format}}(o) = 1$ if the model response contains valid *plan* and *thinking tags* while being not too short, and 0 otherwise. This ensures the model maintains systematic reasoning structure and avoids thinking degradation.

**Policy Generation and Update.** We optimize the policy using GRPO. For each instruction $q$, we sample $N$ responses $\{o_1, \ldots, o_N\}$ from the current policy $\pi_\theta$ and compute their rewards $r_i = R_{\text{format}}(o_i) + R_{\text{grounding}}(o_i)$. We then compute group-normalized advantages $A_i = \frac{r_i - \text{mean}(\{r_1, \ldots, r_N\})}{\text{std}(\{r_1, \ldots, r_N\})}$. The policy $\pi_\theta$ is updated via clipped surrogate objective:

$$\mathcal{L}_{\text{GRPO}}(\theta) = \mathbb{E}_{q, \{o_i\}} \left[ \frac{1}{N} \sum_{i=1}^{N} \left( \min \left( \frac{\pi_\theta(o_i|q)}{\pi_{\theta_{\text{old}}}(o_i|q)} A_i, \right. \right. \right.$$
$$\left. \left. \text{clip} \left( \frac{\pi_\theta(o_i|q)}{\pi_{\theta_{\text{old}}}(o_i|q)}, 1 - \varepsilon, 1 + \varepsilon \right) A_i \right) \right)$$
$$\left. - \beta \mathcal{D}_{\text{KL}}(\pi_\theta \| \pi_{\text{ref}}) \right] \quad (3)$$

### 3.3. Modular Inference Enhancement via Symbolic Decoupling

A unique advantage of APEIRIA unattainable by black-box 3D MLLMs is the explicit decoupling of planning (reasoning) and execution (perception and spatial verification). While standard 3D MLLMs fuse these processes into an opaque latent space, our neuro-symbolic architecture maintains a transparent interface, enabling plug-and-play integration of external state-of-the-art components during inference without retraining.

**External Plan Injection.** Our structural CoT acts as a universal control interface that allows self-generated plans to be seamlessly replaced by external planners. During inference, we can inject refined decomposition plans from frontier LLMs (e.g., GPT-4 (OpenAI, 2023)) into the reasoning trace. This enables our model to function as a *visual execution engine*, leveraging the superior planning capabilities of larger models to solve complex queries that exceed its internal capacity.

**Perception Module Replacement.** Similarly, our framework supports modular upgrades of execution components. As an example, we demonstrate replacing the `scene()` primitive execution which acts as a universal interface for perception models. We replace the default `scene()` exeuction by model itself with outputs from state-of-the art instance segmentation models (e.g., SegDINO3D (Qu et al., 2026)) by simply formatting their outputs to match the CoT trace. This design facilitates continuous improvement of our reasoning system from advancements in 3D perception.

## 4. Experiments

### 4.1. Evaluation and Implementation Details

**Datasets and Metrics.** Following previous NS methods (Hsu et al., 2023; Feng et al., 2024), we evaluate APEIRIA on spatial reasoning 3D vision-language benchmarks, namely ScanRefer (Chen et al., 2020) and Multi3DRefer (Zhang et al., 2023). For ScanRefer, we report accuracy at IoU thresholds of 0.25 and 0.5 (Acc@0.25, Acc@0.5), and F1 scores for Multi3DRefer. To further assess generalizability, we also evaluate our method on SQA3D (Ma et al., 2023) for situated question answering (Exact Match) and Scan2Cap (Chen et al., 2021) for dense captioning (CiDEr@IoU scores).

**Implementation Details.** We implement APEIRIA on a standard 8B MLLM backbone (Yang et al., 2025), employing AdamW (Loshchilov et al., 2019) and Muon (Jordan et al., 2024; Liu et al., 2025) optimizer with LoRA (Hu et al., 2022) for efficient fine-tuning. Please refer to the supplementary for full details.

### 4.2. Performance on 3D Spatial Reasoning

As illustrated in Table 1, compared to state-of-the-art **3D MLLMs** (e.g., Chat-Scene, Inst3D-LMM), APEIRIA achieves superior performance, notably surpassing the strongest baseline on both ScanRefer and Multi3DRefer datasets. This gain empirically demonstrate the effectiveness of our distilled systematic reasoning, which enables more precise spatial verification than the implicit "black-box" logic of standard MLLMs. Conversely, traditional **Neuro-Symbolic methods** (e.g., NS3D) struggle severely on natural language datasets like ScanRefer and Nr3D due to their closed-set vocabulary constraints and rigid program parsers. It fails to generalize to the linguistic diversity of real-world instructions, resulting in subpar performance. These results confirms that APEIRIA could combine the best of both worlds: the systematic reasoning of symbolic programs and the semantic flexibility of LLMs.

### 4.3. Generalization to Open-Set Concepts

Traditional NS methods (Hsu et al., 2023; Feng et al., 2024) are fundamentally limited by their reliance on closed-set vocabularies. Does APEIRIA's distilled reasoning capabilities generalize to open-set concepts unseen during training? To evaluate this, we leverage the pairing of Sr3D (with synthetic instructions) and Nr3D (with natural human language instructions) datasets (Achlioptas et al., 2020), which share identical scenes but differ in linguistic complexity. We train our 3D MLLM *exclusively on the Sr3D* (Stage 2-only) and evaluate it in a **training-free zero-shot** setting on the Nr3D with unseen concepts and instruction structures. This setup strictly tests whether the reasoning patterns distilled from

*Table 1.* **3D Spatial Reasoning Performance Comparison.** We compare methods based on their output format: 'Head' denotes methods using an extra grounding decoder, while 'Text' denotes methods that output box boundaries or identifiers directly as plain text. Best and second best are in **bold** and underlined. [†]: with modular inference enhancement.

| Method | Output | ScanRefer | | Multi3DRefer | |
|---|---|---|---|---|---|
| | | Acc@0.25 | Acc@0.5 | F1@0.25 | F1@0.5 |
| *Neuro-Symbolic Methods* | | | | | |
| NS3D (Hsu et al., 2023) | Head | 22.4 | - | - | - |
| LARC (Feng et al., 2024) | Head | 32.9 | - | - | - |
| LaSP (Mi et al., 2025) | Text | 49.2 | - | - | - |
| *Specialist Methods* | | | | | |
| ScanRefer (Chen et al., 2020) | Head | 37.3 | 24.3 | - | - |
| M3DRef-CLIP (Zhang et al., 2023) | Head | 51.9 | 44.7 | 42.8 | - |
| 3D-VisTA (Zhu et al., 2023) | Head | 50.6 | 45.8 | - | - |
| SceneVerse (Jia et al., 2024) | Head | - | 48.1 | - | - |
| *3D MLLMs* | | | | | |
| Grounded 3D-LLM (Chen et al., 2024c) | Head | 48.6 | 44.0 | 44.7 | 40.8 |
| LLaVA-3D (Zhu et al., 2025) | Head | 50.1 | 42.7 | 49.8 | 43.6 |
| PQ3D (Zhu et al., 2024) | Head | 57.0 | 51.2 | - | 50.1 |
| Video-3D LLM (Zheng et al., 2025) | Head | 58.1 | 51.7 | 58.0 | 52.7 |
| 3D-LLaVA (Deng et al., 2025) | Text | 51.2 | 40.6 | - | - |
| Chat-Scene (Huang et al., 2024a) | Text | 55.5 | 50.2 | 57.1 | 52.4 |
| Inst3D-LMM (Yu et al., 2025) | Text | 57.8 | 51.6 | 58.3 | 53.5 |
| APEIRIA (Ours) | Text | 58.4 | 51.2 | 59.2 | 53.8 |
| APEIRIA[†] (Ours) | Text | **60.5** | **53.2** | **60.9** | **55.2** |

*Table 2.* **Zero-shot Generalization on Open-Set Concepts.** We compare APEIRIA (Zero-shot, never seen Nr3D) against NS methods. Best performance in **bold**.

| Method | Training Data | Setting | Acc |
|---|---|---|---|
| NS3D (Hsu et al., 2023) | Sr3D | Zero-shot | 7.3 |
| NS3D (Hsu et al., 2023) | Nr3D | Supervised | 33.9 |
| R2G (Li et al., 2025b) | Sr3D | Zero-shot | 25.8 |
| **APEIRIA (Ours)** | Sr3D | Zero-shot | **36.5** |

programs can generalize to free-form descriptions that could defy symbolic parsing.

As shown in Table 2, APEIRIA achieves **36.5%** accuracy on Nr3D without seeing any real-world annotations. Remarkably, this zero-shot performance surpasses the **fully supervised** NS3D baseline (trained on Nr3D) by **+2.6%**. While NS3D fails to learn and generalize open-vocabulary concepts (e.g., "messy", "cozy"), APEIRIA leverages the LLM's semantic knowledge to interpret these nuanced descriptors. This empirically validates that our paradigm could break the "vocabulary bottleneck" itraditional neuro-symbolic systems, enabling effective sim-to-real transfer of the reasoning capabilities.

### 4.4. Generalization to More Tasks

To validate that our curriculums is a general training paradigm rather than a grounding-specific pipeline, we further evaluate APEIRIA on two additional 3D scene understanding tasks (i.e., SQA3D (Ma et al., 2023) for **Situated**

**Question Answering** and Scan2Cap (Chen et al., 2021) for **Dense Captioning**) by simply swapping the outcome reward in Stage 3: exact-match (EM) for question answering and CIDEr for dense captioning. No modification to the symbolic reasoning pipeline in Stages 1 and 2 is required. Specifically, SQA3D requires 3D MLLM to conduct perspective-dependent spatial reasoning, answering questions based on the asker's position, and is evaluated by answer exact-match (EM) score. Scan2Cap requires generating spatially grounded natural language descriptions for objects in 3D scenes, evaluated by CiDEr scores conditioned on localization accuracy.

As shown in Table 3, APEIRIA matches or surpasses prior best methods across all metrics on both tasks, despite being primarily designed for spatial grounding, confirming that our three-stage curriculum is a training paradigm generalizable to various 3D scene understanding tasks.

### 4.5. Ablation Studies

We conduct comprehensive ablation studies on ScanRefer and Multi3DRefer to validate our curriculum design, reward formulation, and modular architecture.

**Impact of Curriculum Stages.** We first investigate the necessity of our progressive training pipeline (Table 4). **(1) Efficacy of RL (Stage 3):** Replacing the CoT-RL stage with direct-answer SFT on the same downstream data results in a significant performance drop of 6.9% and 3.9% on ScanRefer and Multi3DRefer, respectively. This confirms that while Stage 2 teaches the "syntax" of reasoning, RL is

*Table 3.* Results on Scan2Cap (dense captioning) and SQA3D (situated question answering). Best and second best are in **bold** and underlined.

| Method | Scan2Cap | | | SQA |
|---|---|---|---|---|
| | ScanRefer | | Nr3D | |
| | C@.25 | C@.5 | C@.5 | EM |
| *Specialist Methods* | | | | |
| Scan2Cap (Chen et al., 2021) | 56.8 | 39.1 | 27.5 | – |
| SQA3D (Ma et al., 2023) | – | – | – | 47.2 |
| 3D-VisTA (Zhu et al., 2023) | – | 61.6 | – | 48.5 |
| V2C-DETR (Chen et al., 2023) | 71.4 | 61.8 | 43.8 | – |
| V2C-DETR++ (Chen et al., 2024b) | 76.4 | 67.6 | 47.1 | – |
| BridgeQA (Mo et al., 2024) | – | – | – | 52.9 |
| PQ3D (Zhu et al., 2024) | – | 80.3 | – | 47.1 |
| *3D MLLMs* | | | | |
| LL3DA (Chen et al., 2024a) | 74.2 | 65.2 | 51.2 | – |
| LEO (Huang et al., 2024b) | – | 72.4 | – | 50.0 |
| SceneLLM (Fu et al., 2025) | – | – | – | 53.6 |
| ChatScene (Huang et al., 2024a) | – | 77.2 | – | 54.6 |
| LEGO (Mo et al., 2025) | 84.7 | 78.6 | 61.4 | 54.8 |
| LLaVA-3D (Zhu et al., 2025) | – | 79.2 | – | 55.6 |
| Inst3D-LMM (Yu et al., 2025) | – | 79.7 | – | – |
| Video-3D LLM (Zheng et al., 2025) | – | 83.8 | – | **58.6** |
| **APEIRIA (Ours)** | **90.6** | **84.1** | **68.1** | **58.6** |

crucial for adapting these patterns to the complex linguistic structures of real-world instructions. **(2) Necessity of Reasoning Injection (Stage 2):** Attempting to train CoT-RL directly from the alignment stage leads to a catastrophic performance degradation. This might due to VLM's lack of inherent knowledge of 3D spatial verification, and without the "warm start" provided by structured reasoning in symbolic programs, the RL agent struggles to explore the reasoning space from scratch effectively.

**Analysis of RL Rewards.** We further dissect the contribution of our reward components with following observations (Table 4): **(1) Format Reward is Critical:** Removing the format constraint causes moderate performance drop with response length degration. Qualitative inspections reveals that without this constraint, the model frequently suffers from "structure collapse" that either only outputting instructions again or skipping the reasoning process entirely, since they are the "shortcut" path to maximize reward at the early RL stage. **(2) Soft Grounding Reward Benefits:** Replacing our continuous configuration-based reward with sparse IoU feedback results in a slight but consistent 0.7% and 0.5% decline. While not the most effective factor, dense feedback helps guide RL optimization.

**Inference Strategy Analysis.** Does the model actually need to "think"? When we force the fully trained model to output the answer directly without generating the thinking trace, performance drops by **1.6%** and **1.0%** (Table 4). Interestingly, this "direct" inference still significantly outperforms the Stage 2 (SFT) baseline (56.8 vs 51.5 and 58.2 vs. 55.3),

*Table 4.* **Design Ablations.** We evaluate the impact of our curriculum stages, RL reward components and whether to think, on ScanRefer and Multi3DRefer datasets. Best performance in **bold**.

| Variant | ScanRefer Acc@0.25 | Multi3DRefer F1@0.25 |
|---|---|---|
| **APEIRIA** | **58.4** | **59.2** |
| *Training Curriculum Stages* | | |
| w/o Stage 3 (No CoT-RL) | 51.5 | 55.3 |
| *(Stage 3 → Direct SFT)* | -6.9 | -3.9 |
| w/o Stage 2 (No CoT-SFT) | 48.2 | 36.7 |
| *(Direct Alignment → CoT-RL)* | -10.2 | -22.5 |
| *RL Reward Components* | | |
| w/o Format Reward | 55.7 | 57.1 |
| *(Grounding Reward Only)* | -2.7 | -2.1 |
| w/o Soft Grounding Reward | 57.7 | 58.7 |
| *(Use IoU Reward Instead)* | -0.7 | -0.5 |
| *Inference Strategy* | | |
| w/o Thinking | 56.8 | 58.2 |
| *(Direct Answer Output)* | -1.6 | -1.0 |

*Table 5.* **RL Benefit by Reasoning Complexity** on ScanRefer (Acc@0.5). CoT-RL yields larger gains on longer, more complex reasoning chains where Stage 2 CoT-SFT lacks supervised traces.

| Complexity | Steps | SFT-only | CoT-RL | Δ |
|---|---|---|---|---|
| Short | ≤4 | 47.2 | 45.4 | −1.8 |
| Medium | =5 | 50.7 | 52.2 | +1.5 |
| Long | ≥6 | 45.2 | 47.9 | +2.7 |

suggesting that RL has improved the model's internal representations, but the explicit thinking procedure remains essential for unlocking its full spatial reasoning potential.

**Impact of Reasoning Complexity on RL Benefit.** To understand *where* Stage 3 (CoT-RL Stage) contributes most, we group ScanRefer validation samples by reasoning complexity, estimated by the number of plan steps in the generated CoT, and compare models trained with and without RL. As shown in Table 5, a clear trend emerges: **RL benefit increases with reasoning complexity.** For simple queries (≤4 steps), SFT from synthetic programs already captures sufficient patterns, and RL exploration introduces marginal noise. For complex queries requiring multi-step spatial verification (≥6 steps), CoT-RL provides consistent improvement (+2.7%). This is because CoT-SFT lacks supervised traces for longer programs, while Stage 3's outcome-based RL can reward correct longer traces discovered through exploration, effectively providing supervision signals for complex reasoning chains that CoT-SFT cannot cover. This further validates our Stage 3's role: extending systematic neuro-symbolic reasoning to complex, real-world instructions where dense symbolic supervision is unavailable.

**Modularity and Scalability.** Finally, we analyze the source of our model's performance via module replacement (Table 6). **(1) Planning Saturation:** Replacing self-generated plans with those from a powerful external LLM (Claude 4.5

*Table 6.* **Modularity Analysis.** We demonstrate the modularity of APEIRIA by replacing its planning and perception executions at inference time. Best non-oracle results are in **bold**. $^\star$: M3DRef stands for Multi3DRefer. $^\dagger$: with modular inference enhancement.

| Module | Source | ScanRefer Acc@0.25 | M3DRef$^\star$ F1@0.25 |
|---|---|---|---|
| **APEIRIA** | | 58.4 | 59.2 |
| *(Self-Plan + Self-Perception)* | | | |
| **Planning** | External Planner | 58.6 | 59.5 |
| | *(Self-Plan $\rightarrow$ Claude 4.5 Opus)* | **+0.2** | **+0.3** |
| **Perception** | Mask3D (Schult et al., 2023) | 58.5 | 58.3 |
| | *(External Perception)* | **+0.1** | -0.9 |
| | SegDINO3D (Qu et al., 2026) | 60.4 | 60.6 |
| | *(Strong External Perception)* | **+2.0** | **+1.4** |
| | Oracle (Ground-Truth Labels) | 61.3 | 61.3 |
| | *(Upper Bound)* | +2.9 | +2.1 |
| **APEIRIA**$^\dagger$ | | **60.5** | **60.9** |
| *(Full Modular Enhancement)* | | **+2.1** | **+1.7** |

Opus (Anthropic, 2025)) yields only marginal gains. This indicates that our distilled planner is already highly robust, effectively leveraging the inherent instruction parsing capabilities of the base LLM. **(2) Perception Scalability:** In contrast, replacing the perception module with a stronger semantic recognition model (SegDINO3D (Qu et al., 2026)) significantly boosts performances, nearly matching the Oracle upper bound. This demonstrates that visual perception accuracy limit our current system more than planning failures. However, this limitation effectively highlights the *scalability* of our modular architecture: unlike black-box end-to-end models, APEIRIA allows us to address this visual bottleneck by seamlessly integrating stronger 3D perception models as they emerge without re-training.

### 4.6. Qualitatives of Emergent Reasoning Behaviors

A key hypothesis of our curriculum is that CoT-RL (Stage 3) enables generalization beyond the fixed templates of Stage 2. As shown in Figure 4, qualitative analysis reveals that APEIRIA spontaneously **invents new reasoning patterns** to bridge the gap between symbolic rigor and linguistic flexibility. First, it breaks the "vocabulary bottleneck" by adapting the `filter` primitive to handle **open-vocabulary descriptors** (e.g., generating `filter(beige chair)` instead of rigid categories), effectively leveraging the VLM's semantic knowledge. Second, the model successfully constructs plans for more **deeply nested queries** unseen during SFT. To resolve multi-condition constraints (e.g., "next to A **AND** left of B"), it spontaneously emerges new logical primitives such as `intersection` and `union`, which are never explicitly taught in the curriculum. This indicates that APEIRIA has not merely memorized program templates but has internalized the underlying "syntax" of spatial logic, allowing it to dynamically compose novel primitives to match

[think]
I need to find the object described as: "this **beige chair** is next to the coat rack. it is also to the left of the table and lamp."
*// Planning*
Let's plan my next steps: Find objects of category **'beige chair'**, 'coat rack', 'table', and 'lamp'; Check which **beige chair(s) are 'next to' the coat rack(s) and 'to the left of' the table(s) and lamp(s)**.
*// Execute scene()*
I see 33 object(s) in the scene: 0 (couch), ... [details omitted]
*// Execute filter(beige chair)*
Looking for 'beige chair'. I find 2 object(s): 3, 5, at ...
*// Execute filter(coat rack)*
Looking for 'coat rack'. I find 1 object(s): 17 at...
*// Execute filter(table)*
Looking for 'table'. I find 6 object(s): 7, 8, 9, ...
*// Execute filter(lamp)*
Looking for 'lamp'. I find 3 object(s): 16, 25, 31, at ...
*// Execute relate(filter(beige chair), union(filter(table), filter(lamp)), next to)*
Analyzing... Object 5 is next to Object 17 and to the left of Object 14 and 16. After analysis, I find 1 object(s) 'next to' ...
*// Execute relate(filter(beige chair), filter(coat rack), left)*
After analysis, I find 1 object(s) 'to the left of'...
*// Execute intersection(relate(...), relate(...))*
Now, I'll check which beige chair(s) are 'next to' the coat rack(s) and 'to the left of' the table(s) and lamp(s): **Object 5 (chair)** matches both relations.
[answer]
I've located 1 object(s) as described: Object 5: Coordinates (3.47, 3.18, 2.49), dimensions 0.80 x 0.51 x 1.47

*Figure 4.* Example CoT revealing emerging reasoning patterns in CoT-RL stage. Object location details are omitted for brevity.

the complexity of real-world instructions.

## 5. Conclusion

We presented APEIRIA, a neuro-symbolic 3D MLLM that bridges the gap between interpretable but closed-set symbolic methods and flexible but opaque end-to-end MLLMs. Our key insight is that reasoning patterns can be distilled from symbolic programs into natural language chain-of-thought and integrated into MLLM, enabling systematic spatial reasoning with open-vocabulary flexibility. Through a three-stage curriculum progressing from perception alignment and symbolic reasoning injection to RL-based generalization, APEIRIA learns to decompose queries, verify spatial relations step-by-step, and adapt to complex real-world instructions. Experiments demonstrate that APEIRIA surpasses prior neuro-symbolic methods and matches state-of-the-art 3D MLLMs, while preserving key virtues: transparent reasoning traces for interpretability and modular architecture for scalable upgrades. We hope this work offers a initial stepstone toward interpretable and useful embodied agents.

## Impact Statement

This work advances interpretable 3D spatial reasoning through neuro-symbolic Chain-of-Thought distillation. Our framework seeks to establish a more systematic and verifiable grounding of objects within 3D environments, offering potential societal benefits for assistive robotics, indoor navigation, and embodied AI systems, where transparent failure analysis is critical for trust calibration, safety, and effective debugging.

## Acknowledgment

This work was supported by the grants from the Beijing Natural Science Foundation 4252040, Beijing Nova Program and National Natural Science Foundation of China 62372014 and CAAI-Tencent Rhino-Bird Open Research Fund.

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

# A. Supplementary Material

In this supplementary material, we provide comprehensive details to support reproducibility and further understanding of our method. Section A.1 presents implementation details including detailed model architecture specifications and training protocols for all three curriculum stages. Section A.2 elaborates on the data construction process: the perception alignment mixture (Stage 1), the program-to-CoT translation pipeline with verified execution traces (Stage 2), and the RL training configuration (Stage 3). Table 7 and Figure 5 provide the instruction templates and a complete reasoning trace example used throughout training and evaluation. Section A.4 presents additional experiments on backbone scaling and reasoning trace quality analysis. Section A.5 further explains how modular inference enhancement works with concrete examples. Finally, Section A.6 discusses limitations of our approach.

## A.1. Implementation Details

**Model Architecture.** We implement APEIRIA on Qwen3-VL-Instruct (Yang et al., 2025), a state-of-the-art 8B parameter multi-modal large language model. For 3D scene encoding, we follow prior work (Huang et al., 2024a) to segment input scenes using Mask3D (Schult et al., 2023) and extract visual features using Uni3D (Zhou et al., 2024) for 3D geometric information and DINOv2 (Oquab et al., 2024) for visual appearance features.

For object spatial encoding, we adopt a lattice-based positional encoding that represents each object's 6D configuration (location $x, y, z$ and size $h, w, l$) as a continuous embedding. Specifically, for each spatial axis $a \in \{x, y, z, h, w, l\}$, we define a learnable embedding matrix $\boldsymbol{H}^a \in \mathbb{R}^{N \times d}$, where $N = 200$ is the number of uniformly spaced grid points along the axis. Given an object with normalized configuration $\boldsymbol{p} = (x, y, z, h, w, l) \in [0, 1]^6$, we compute the positional embedding as:

$$\boldsymbol{h}_{\text{pos}}(\boldsymbol{p}) = \sum_{a \in \{x,y,z,h,w,l\}} \sum_{k=1}^{N} w_k^a \cdot \boldsymbol{H}_k^a, \quad \text{where} \quad w_k^a = \exp\left(-\left(N \cdot p_a - k\right)^2\right) \tag{4}$$

Intuitively, the Gaussian weights $w_k^a$ assign higher influence to grid points closer to the object's actual coordinate, producing a smooth interpolation over the learnable lattice embeddings. The final object representation concatenates this spatial embedding with visual features and is interleaved with instruction tokens as LLM input.

**Training Protocol.** We employ LoRA (Hu et al., 2022) with rank $r = 128$ for parameter-efficient fine-tuning across all stages. Stage 1 (Perception Alignment) trains for 3 epochs on our alignment data mixture with a learning rate of $1 \times 10^{-5}$ and batch size of 16. Stage 2 (CoT-SFT) trains for 1 epoch on the program-derived reasoning traces with a learning rate of $1 \times 10^{-5}$ and batch size of 8. Stage 3 (CoT-RL) employs Group Relative Policy Optimization (GRPO) (Shao et al., 2024) with a group size of $G = 8$ rollouts per prompt, trained for 1 epoch with a learning rate of $3 \times 10^{-6}$ with a batch size of 128 prompts (1024 rollouts). We set $\varepsilon = 0.2$ for GRPO clipping and $\beta = 0$ as KL divergence coefficient to further reduce GPU memory consumption and enhance exploration. All experiments use AdamW (Loshchilov et al., 2019) optimizer with weight decay of 0.001 and are conducted on 8 NVIDIA A100-40G GPUs.

A significant engineering challenge for Stage 3 is the lack of native 3D multi-modal support in existing RL libraries. To address this, we developed a custom infrastructure built upon SGLang (Zheng et al., 2024) that enables efficient policy rollouts with arbitrary 3D object-centric multi-modal features. Our implementation combines acceleration techniques from SGLang and achieves approximately 3× speedup compared to naive generation with huggingface's `transformers` library, making large-scale 3D RL exploration and training tractable.

## A.2. Data Construction Details

**Stage 1: Perception Alignment.** To establish robust 3D scene understanding capabilities, we first construct a balanced mixture of object-centric tasks. These tasks include

- **Object Recognition:** Tasks include category ("What is this object?") and attribute ("What attribute does this object have?") identification, targeting only single-object feature inputs. We collect object category annotations from ScanNet (Dai et al., 2017) and attribute annotations from MMScan (Lyu et al., 2024). Sourced from ScanNet and textual attributes extracted from captions.

- **Object Localization:** Tasks involve outputting 3D coordinates of specified objects based on its object ID ("Where is

the object with ID `<id>` located?"). We utilize ground-truth bounding box centers from ScanNet (Dai et al., 2017) as supervision.

- **Object Captioning:** Generating one-sentence descriptions for single objects using ScanRefer (Chen et al., 2020) and ReferIt3D (Achlioptas et al., 2020) captions where the target is unambiguous.

The total alignment dataset comprises approximately 193K instruction-response pairs.

**Stage 2: Symbolic Reasoning Injection via CoT-SFT.** For CoT-SFT, we translate neuro-symbolic programs into iterative natural language reasoning traces. Each program step is converted into a **Plan** sentence describing the operation, and an **Execution** block detailing the intermediate results. We build the final CoT by first concatenating all step-by-step Plans and then all Executions, followed by a simple summarization into the final answer. To enable thinking, we simply add a sentence "Think about the scene first." in the instruction prompt sent to model during CoT-SFT, which is also perserved at CoT-RL stage.

For Level 1 programs, we construct single-step `filter` operations from ScanNet category annotations and MMScan attribute annotations, yielding approximately 78K traces with verified outputs. For Level 2 programs, we leverage the Sr3D dataset (Achlioptas et al., 2020), which provides synthetic instructions guaranteed to be solvable in exactly two steps. Each instruction maps to a two-step program (e.g., `relate(filter(A), filter(B), r)`), and we construct complete execution traces by verifying intermediate `filter` outputs against object annotations and final `relate` outputs against target annotations. This yields approximately 66K traces with perfect step-by-step supervision. The specific templates used are as follows:

- **Plan Generation:**
  - `filter(C)` → "Find all objects of category [C]."
  - `relate(A, B, r)` → "Check which [A] are [r] to [B]."

- **Execution Generation:**
  - `scene()` → I see [$N$] objects in the scene: Object 1: [$Category_1$], Object 2: [$Category_2$], ...
  - `filter()` → Looking for [Category]... I found [$k$] objects: Object [$ID_1$]: at [Position], Object [$ID_2$]: at [Position], ...
  - `relate()` → Analyzing relation [r]... Object [$ID_A$] is [r] to Object [$ID_B$], Object [$ID_C$] is [not r] to Object [$ID_D$]...

The full algorithm for building CoT data from programs is summarized in Algorithm 1, and a full example of the assembled CoT is shown in Figure 5.

**Stage 3: Open-Set and Complex Reasoning Generalization via CoT-RL.** For reinforcement learning, we use the training splits of ScanRefer (Chen et al., 2020) and Multi3DRefer (Zhang et al., 2023), which contain natural language instructions with complex nested structures and open-vocabulary concepts. Since ground-truth execution traces are unavailable for these datasets, we rely solely on outcome supervision (whether the predicted bounding box matches the target) combined with our format reward to guide exploration.

### A.3. Instruction Prompts

Table 7 summarizes the instruction and response templates used in APEIRIA. In scene context prompts, `|object_set|` placeholder will be replaced by real object-centric scene features. Object positions and dimensions are formatted with 2 decimal places precision.

### A.4. Additional Experiments

**Backbone Scaling.** To evaluate the impact of the underlying MLLM backbone size on 3D spatial reasoning performance, we conduct experiments with different model scales within our APEIRIA framework. As shown in Table 8, scaling up from

---

**Algorithm 1:** Building CoT Data from Programs

---

**Data:** Program $P$, Ground truth annotations $\mathcal{A}$
**Result:** Chain-of-Thought data with plans **Plans** and executions **Execs**
```
// Parse program into execution sequence
```
$\mathcal{T} \leftarrow$ PARSEAST($P$)
$\mathcal{S} \leftarrow$ GETEXECUTIONSEQUENCE($\mathcal{T}$)
```
// Initialize plan and execution lists
```
**Plans** $\leftarrow []$
**Execs** $\leftarrow []$
```
// Process each execution step
```
**foreach** *statement $s$ in $\mathcal{S}$* **do**
  **if** $s$ *is* `scene()` **then**
    $\mathcal{O}_{in} \leftarrow \emptyset$
    $\mathcal{O}_{out} \leftarrow$ all objects in scene
  **else if** $s$ *is* `filter(condition)` **then**
    $\mathcal{O}_{in} \leftarrow \mathcal{O}$
    $\mathcal{O}_{out} \leftarrow \{o \in \mathcal{O} \mid o \text{ matches } condition\}$
  **else if** $s$ *is* `relate(...)` *or* `relate_anchor(...)` *or* `relate_triple(...)` **then**
    $\mathcal{O}_{in} \leftarrow$ filtered object sets from previous steps
    $\mathcal{O}_{out} \leftarrow$ target objects from $\mathcal{A}$
  `// Generate plan and execution for this step`
  $plan \leftarrow$ PROGRAMTOPLAN($s$)
  $exec \leftarrow$ FORMATEXECUTION($s, \mathcal{O}_{in}, \mathcal{O}_{out}$)
  **Plans**.APPEND($plan$)
  **Execs**.APPEND($exec$)
**end**
```
// Concatenate plans and executions
```
$CoT_{plans} \leftarrow$ CONCATENATE(**Plans**)
$CoT_{execs} \leftarrow$ CONCATENATE(**Execs**)
**return** CONCATENATE($CoT_{plans}, CoT_{execs}$)

---

*Table 7.* **Prompt Templates.** Representative templates for scene context, instructions, object formatting, and responses used in APEIRIA.

| Component | Template |
|---|---|
| Scene Context | `These are all objects in the scene: |object_set|` |
| Instruction | `Think about the scene first. Identify the object described as:` `"{description}". Respond with the object's ID, position, and size.` |
| Object Locations | `Object {id}: {name} at ({x}, {y}, {z}), size: {w} x {h} x {d}` |
| Response | `I found {count} object(s) matching the description: {object_locations}` |
| Chain-of-Thoughts | `[think] {thinking procedure...} [answer]` |

4B to 8B parameters yields consistent performance improvements across all metrics, demonstrating the benefits of larger model capacity for 3D vision-language understanding.

**Reasoning-Trace Quality Analysis.** A natural concern is whether outcome-RL in Stage 3 (CoT-RL Stage) preserves the structured reasoning patterns injected in Stage 2 (CoT-SFT), or degrades them into reward-hacking shortcuts. We then conduct a systematic quality analysis on 200 randomly sampled reasoning traces from the ScanRefer validation set after Stage 3 training.

Specifically, we use Claude Opus 4.6 (Anthropic, 2025) as an automated judge, evaluating each trace along two criteria:

- **Neuro-Symbolic (NS-) Compliance:** Whether the trace is parsable into a valid symbolic program structure (i.e., a well-formed sequential combination of primitives like `scene`, `filter` and `relate`).
- **Reference Consistency:** Whether objects are consistently referenced (same ID ↔ same entity) throughout the chain, without contradictory or hallucinated references.

We validate the automated judge on 50 manually annotated traces and observe **94% agreement** with human annotations,

These are all objects in the scene: |object_set|. Think about the scene first. Find the vase left to the computer. Provide its ID, position, and dimensions.

[think]

I need to find the vase left to the computer.

*// Stepwise plans.*

Let's plan my next steps: Examine all objects in the scene; Find all objects of category 'vase'; Find all objects of category 'computer'; Check which vase(s) are 'left to' the computer.

*// Execution of scene() statement.*

First, let me list the object IDs and names: Object 0(vase), 1(vase), 2(bottle), 3(rug), ... [others omitted]

*// Execution of filter() statement.*

Among these objects, I can see the vase(s):

Object 0: At (1.15, 6.09, 1.33), size: 0.86 x 0.99 x 1.79

Object 1: At (6.66, 5.41, 0.29), size: 1.60 x 1.09 x 1.70

And the computer(s):

Object 4: At (2.34, 3.50, 0.75), size: 1.20 x 0.60 x 1.10

Object 11: At (1.44, 4.04, 3.10), size: 0.87 x 1.38 x 0.87

*// Execution of relate() statement.*

Now, I will analyze if any vase is left to the computer:

The vase (ID 0) is left to the computer (ID 4).

The vase (ID 0) is above and near the computer (ID 11).

The vase (ID 1) is right to and near the computer (ID 4).

The vase (ID 1) is right to, in front of and far from the computer (ID 11).

*// Summarization.*

From analysis above, the vase (ID 0) is left to a computer (ID 4). Now, I will formulate the response based on the identified objects.

[answer]

I found the vase left to the computer: Object 0: At (1.15, 6.09, 1.33), size: 0.86 x 0.99 x 1.79

*Figure 5.* A full example of assembled thinking trace for APEIRIA. The trace includes the scene context, instruction, step-by-step thinking trace with plans and executions, and the final answer.

*Table 8.* **Backbone Scaling.** Performance comparison of APEIRIA with different backbone sizes. Larger backbones yield consistent improvements.

| Backbone | ScanRefer | | Multi3DRefer | |
|---|---|---|---|---|
| | Acc@0.25 | Acc@0.5 | F1@0.25 | F1@0.5 |
| Qwen3-VL-4B | 56.4 | 48.9 | 56.1 | 51.0 |
| Qwen3-VL-8B | **58.4** | **51.2** | **59.2** | **53.8** |
| Δ (4B → 8B) | +2.0 | +2.3 | +3.1 | +2.8 |

confirming its reliability.

As shown in Table 9, two findings emerge. **(a)** RL preserves structured reasoning: 92.5% of post-RL traces remain parsable into symbolic programs, demonstrating that outcome-only RL does not collapse the reasoning structure injected in Stage 2, even though the predicates are now open-vocabulary natural language that goes beyond what traditional NS3D frameworks can express. **(b)** Reasoning quality correlates with answer correctness: correctly grounded samples show **+13.4%** higher joint compliance than incorrect ones, indicating that the model's grounding accuracy is linked to its reasoning quality rather than memorized shortcuts. These results confirm that our curriculum could successfully internalize faithful, verifiable reasoning patterns that persist through RL optimization and directly contribute to downstream performance.

*Table 9.* **Reasoning-trace quality analysis**. Correctly grounded samples exhibit substantially higher structural compliance than incorrect ones, indicating that structured reasoning directly contributes to accuracy.

| Criterion | Overall | In Correct Samples | In Incorrect Samples | Δ |
|---|---|---|---|---|
| NS-Compliance | 92.5 | 95.3 | 90.4 | +4.9 |
| Reference Consistency | 86.5 | 91.8 | 82.6 | +9.2 |
| Both criteria met | 80.5 | 88.2 | 74.8 | +13.4 |

## A.5. Modular Inference Enhancement Example

Figure 5 illustrates a complete trace where the model generates all four thinking phases (planning, scene perception, execution, answer) end-to-end without external modules. We further provide a concrete example to demonstrate the modular replacement mechanism described in Section 3.3, based on an examplar query *"find the chair to the left of the desk near the window."*

**External Plan Injection.** As shown in Figure 6a, the Planning block can be replaced by an external planner (e.g., Claude Opus 4.5 (Anthropic, 2025)) before decoding begins. The model then decodes all subsequent phases normally while conditioned on the injected plan.

**Perception Module Replacement.** Figure 6b further illustrates the case where the model decodes the Planning block normally, after which the `scene()` output is replaced by a stronger 3D perception model (e.g., SegDINO3D (Qu et al., 2026)). The model continues decoding and benefit from the more accurate object list.

In both replacement cases (Figures 6a and 6b), the model simply conditions on the injected text as if it had generated it itself, and continues decoding the remaining phases without any modification or retraining.

---

[think]
I need to find: "the chair to the left of the desk near the window."
*// Planning (E.g., injected from Claude Opus 4.5)*
Let's plan: First find all 'window'; Then find 'desk' that are 'near' those windows; Finally find 'chair' that are 'to the left of' those desks.
*// Execute scene, filter, relate ...*
*(model decodes normally from here) ...*
[answer]
*(model answers normally)*

---

*(a)* External plan injection. Blue text is injected from an external planner; the model continues decoding the remaining phases.

---

[think]
I need to find: "the chair to the left of the desk near the window."
*// Planning*
Let's plan: Find objects of category 'chair', 'desk', and 'window'; ...
*(model decodes normally up to here)*
*// Execute scene() (E.g., injected from SegDINO3D)*
I see 33 object(s) in the scene: 0 (sofa), 1 (armchair), 2 (desk), 3 (window), 4 (floor lamp), ... [more objects omitted]
*// Execute filter, relate ...*
*(model decodes normally, benefiting from more accurate perception) ...*
[answer]
*(model answers normally)*

---

*(b)* Perception module replacement. Green text is injected from an external detector; the model continues decoding after the enhanced perception.

*Figure 6.* Modular inference enhancement examples.

## A.6. Limitations

While APEIRIA demonstrates strong performance on 3D spatial reasoning benchmarks, several limitations merit discussion.

**Perception Bottleneck.** As evidenced by our modularity analysis (**Modularity and Scalability.** in Section 4.5), the system's overall performance remains fundamentally bounded by the accuracy of the underlying 3D perception module; when object detection or semantic recognition fails, the reasoning engine cannot recover regardless of its logical correctness, as it operates on incomplete or erroneous scene representations.

**Deep Reasoning Chains.** Although our method exhibits emergent composition of logical primitives such as `intersection` and `union` for multi-condition constraints, the reliability and coherence of very deep reasoning chains

involving five or more nested operations have not been extensively validated, and such complex traces may be susceptible to error accumulation during RL optimization and is rare in current datasets.

**Physics and Dynamics Reasoning.** Our symbolic schema targets compositional spatial reasoning over static scene geometry. Queries that require understanding of physical dynamics or causal simulation (e.g., *"which supporting furniture would cause the shelf to collapse if removed?"*) fall outside the current primitive set, as they demand commonsense reasoning about forces, stability, and temporal state changes that are complementary to the spatial decomposition our framework provides. Extending the primitive vocabulary to include physics-aware predicates is a promising future direction.

**Generalization to Outdoor and Navigation Domains.** Our experiments focus exclusively on indoor scenes from ScanNet-family benchmarks. While the core reasoning primitives (`scene`, `filter`, `relate`) are domain-general in principle, we have not empirically validated this transfer. Domain adaptation would likely require extending the primitive set with domain-specific predicates (e.g., traversability, agent dynamics) and constructing seed program traces from outdoor datasets with object pose annotations. We consider this a valuable direction for future work.

