# OpenReview forum: "Distilling Neuro-Symbolic Programs into 3D Multi-modal LLMs"
_ICML.cc/2026/Conference — ICML 2026 regular_

### Official Review · Reviewer_7qpB · 2026-03-04

**Soundness:** 3
**Presentation:** 3
**Significance:** 3
**Originality:** 3
**Overall Recommendation:** 4
**Confidence:** 4

**Summary:**

This paper proposes a chain-of-thought (CoT) enhanced 3D vision-language model by distilling structured neuro-symbolic reasoning into a 3D VLM. Specifically, the authors first construct a neural-symbolic reasoning framework that converts 3D scene understanding tasks into structured, formatted reasoning traces, enabling explicit intermediate logical steps grounded in 3D spatial representations. These formatted reasoning texts are then used to supervise the training of a 3D vision-language model following a reasoning-oriented paradigm: the model is first fine-tuned with CoT-style supervised learning (CoT-SFT) and subsequently optimized using reinforcement learning (GRPO) to further improve reasoning consistency and answer correctness. Through this two-stage training pipeline, the model learns to generate interpretable reasoning chains for 3D spatial tasks. Experiments on multiple 3D reasoning and embodied understanding benchmarks demonstrate improved reasoning accuracy and enhanced generalization compared to standard 3D VLM training strategies.

**Compliance With Llm Reviewing Policy:**

Affirmed.

**Final Justification:**

This is overall a good paper with a clear presentation, and I choose to keep the positive score after the rebuttal phase. During the rebuttal, my questions about the neural symbolic dataset and the implementation were resolved. I think this paper fits the requirements of ICML and suggests a score of 4.

**Key Questions For Authors:**

The questions below are corresponding to the weaknesses.
1. Can we phase all 3d-spatial tasks in format of symbolics reasoning in this paper? If no, what will be the failure case.
2. How can we constructs a large number of reasoning traces in symbolic format, especially in a new scene such as outdoor navigation tasks.
3. Is Modular Inference Enhancement appliable to traditional Neuro-symbolic methods or specialist methods?

**Limitations:**

yes

**Strengths And Weaknesses:**

**Strengths**

1. Structured reasoning for 3D understanding.
The paper introduces a neuro-symbolic framework that incorporates structured intermediate reasoning steps into 3D vision-language modeling. This design explicitly bridges symbolic scene representations with neural generative models, providing a principled way to inject structured reasoning into 3D understanding.

2. Interpretable and explainable reasoning traces.
By supervising the model with formatted reasoning chains grounded in scene graphs, the approach enables interpretable multi-step explanations for 3D spatial tasks. This improves transparency compared to standard end-to-end 3D VLM training.

3. Clear and well-organized presentation.
The paper is clearly written, with a well-motivated problem formulation and a logically structured description of the training pipeline (CoT-SFT followed by GRPO), making the overall methodology easy to follow.

**Weakness**
1. Limited expressivity of the symbolic 3D representation.
The proposed neuro-symbolic supervision is built upon the Symbolic-3D representation described in Appendix A, which appears to contain a relatively limited set of predefined spatial relations and attributes. Although the vocabulary is open-ended at the language level, the underlying symbolic schema seems restricted to a fixed relational set. This may constrain the expressive capacity of the reasoning space and limit the model to graph-level relational reasoning rather than richer geometric, continuous, or physics-aware 3D reasoning.

2. Scalability and dependence on structured symbolic annotations.
The structured reasoning traces are derived from existing symbolic 3D scene representations and reformatted into chain-of-thought style supervision. While this ensures correctness and interpretability, it raises questions about scalability and generalization to new domains. In contrast to natural language CoT data (which is already scarce in domains like mathematical reasoning), the proposed approach appears to rely on the availability of high-quality symbolic scene graphs. It is unclear how the framework would extend to settings where such structured representations are unavailable or noisy.

3. Modular Inference Enhancement and external model reliance.
The Modular Inference Enhancement stage leverages additional model capabilities during inference. While this improves performance, it introduces ambiguity regarding where the reasoning capability truly resides. Although 3D-MLLMs may not directly support plug-and-play integration of such modules, it remains unclear whether comparable enhancements could also benefit other baseline architectures. A more controlled comparison would help disentangle improvements due to structured reasoning training from gains obtained via external modular components.

---

> ### Author Rebuttal · Authors · 2026-03-31
>
> ### **W1, Q1. Limited expressivity of symbolic 3D representation.**
>
> We acknowledge that our symbolic schema covers a finite set of spatial
> primitives, and certain reasoning types (e.g., physical stability or
> dynamics) are not explicitly captured. However, we argue that our
> framework demonstrates generalization *beyond* the predefined schema:
>
> Our SQA3D experiments (58.6 EM, matching prior best; full results in the
> response to Reviewer arig-W1) provide direct evidence: SQA3D requires
> *perspective-dependent* spatial reasoning (e.g., "what is at my 2
> o'clock?") and continuous angular understanding (e.g., "which object is
> 30 degrees to my left?"). These spatial relations are fundamentally
> different from the discrete relation primitives (left/right, etc.) in
> our Stage 2 symbolic training. The model's strong performance suggests
> that our curriculum distills *general reasoning patterns* (decompose
> query, verify spatial relations) that transfer to novel relation types
> beyond the original symbolic schema, by composing internalized patterns
> with the LLM's numerical and commonsense knowledge.
>
> Regarding the failure cases that our symbolic reasoning are unable to
> cover: we identify **physics-based reasoning** as an important category
> beyond our current scope (e.g., "which supporting furniture will
> collapse the shelf if removed?"). Such queries require commonsense
> reasoning about physical dynamics and simulation over physical world,
> which is complementary to the compositional spatial reasoning our
> framework targets. Extending to physics-aware primitives is a valuable
> future direction.
>
> ### **W2, Q2. Scalability of structured symbolic annotations.**
>
> Our key design principle is that symbolic supervision serves only as a
> *seed* for cold-starting reasoning capability: Stage 2 bootstraps
> reasoning patterns from synthetic programs, then Stage 3 extends them
> via outcome-only RL exploration without any symbolic annotations. Our
> cross-task transfer to more 3D scene understanding tasks and datasets
> (SQA3D, Scan2Cap; please refer to the response to Reviewer arig-W1 for
> experiment details) empirically validates this generalization.
>
> Regarding extension to new domains such as outdoor navigation: we
> acknowledge this is beyond our current scope. However, core spatial
> primitives (proximity, direction) and reasoning primitives (`scene`,
> `filter`, `relate`) are domain-general. To illustrate, consider a real
> query from Talk2Car \[Deruyttere et al., EMNLP 2019\], a referring
> expression dataset built on nuScenes driving scenes:
>
> - **Query**: "Yeah that would be my son on the stairs next to the bus.
>   Pick him up please."
>
> - **Symbolic decomposition**:
>
>       scene()         → enumerate detected objects/agents
>       filter(bus)     → locate buses
>       filter(stairs)  → locate stairs  [new domain concept]
>       relate(next_to, stairs, bus) → stairs near the bus
>       filter(person)  → locate people
>       relate(on, person, stairs)   → find person on those stairs
>
> The reasoning primitives (`relate`, `filter`) transfer directly. Domain
> adaptation would require extending the primitive set with
> domain-specific new predicates if needed (e.g., `stairs`,
> `filter(some action)` for dynamic agents, `traversability` for path
> planning) and constructing initial program traces from datasets with
> object pose annotations. Stage 3 RL could then generalize these patterns
> without additional symbolic annotation. We believe this is a promising
> future direction.
>
> ### **W3, Q3. Modular Inference Enhancement and external model reliance.**
>
> We appreciate this question. We first note that Table 1
> reports results both with and without modular enhancement, and our model
> *without* enhancement already matches/surpasses prior methods.
> Importantly, we do not view modular enhancement as an external add-on,
> but as an *inherent advantage* of our neuro-symbolic design: just as
> modularity is a core merit of traditional NS3D methods, our framework
> brings this same capability to the 3D MLLM through structured CoT
> interfaces.
>
> Concretely, our structured CoT exposes a non-rigorous, natural-language interface
> between reasoning stages. E.g., external model can be used to replace `scene()` (scene perception) execution. The model then continues reasoning over the improved executioon
> without any retraining or architectural change.
>
> This is fundamentally difficult for prior paradigms. NS3D methods
> communicate between modules through jointly trained feature spaces;
> swapping the perception backbone requires retraining all downstream
> concept learners due to feature distribution mismatch. Their program
> parsers are also restricted to fixed grammars, preventing incorporation
> of plans from external LLMs with novel predicates. Standard 3D MLLMs or specialist models lack structured intermediate representations entirely, offering no
> interface for suchj replacement.
> Therefore, this modular capability is a unique merit of our method, combining
> symbolic structure with MLLM flexibility.

---

> > ### Author Rebuttal · Reviewer_7qpB · 2026-04-02
> >
> > See comments, I'll keep the positive score

---

> > > ### Author Response · Authors · 2026-04-07
> > >
> > > Dear Reviewer 7qpB,
> > >
> > > We sincerely appreciate your insightful feedback, which has helped us clarify and strengthen our work.
> > > We appreciate your recognition of our structured and interpretable reasoning design, as well as the clear presentation.
> > > We are glad that we have fully addressed your concerns and will include the new results on the reasoning generalizability and more discussions regarding the unique advantages and boundaries of our method.
> > >
> > > Best regards,
> > >
> > > Authors

---

### Official Review · Reviewer_A5U3 · 2026-03-11

**Soundness:** 3
**Presentation:** 2
**Significance:** 2
**Originality:** 3
**Overall Recommendation:** 4
**Confidence:** 3

**Summary:**

This paper introduces APEIRIA, a framework that enforces planning and execution in neuro-symbolic style on MLLM thinking process. APEIRIA uses a three-stage pipeline that gradually enforces object-centric understanding, neuro-symbolic enforced CoT and complex reasoning. Extensive experiments on ScanRefer, Multi3DRefer, Nr3D and Sr3D and ablation studies of the different components of the pipeline, including training stages and modules, have demonstrated the robustness and efficacy of the proposed framework.

**Compliance With Llm Reviewing Policy:**

Affirmed.

**Final Justification:**

I believe the additional experiments and clarification makes the work more solid. I'll maintain my positive score.

**Key Questions For Authors:**

Please see the Weakness section. In addition, I would prefer an explicit statement of what training data are used for each stage in the main text instead of the appendix.

**Limitations:**

1. Since the method distills neuro-symbolic style reasoning into CoT of the LLM, how strict the LLM will follow such pattern remains unclear.

**Strengths And Weaknesses:**

Strengths:

1. The paper elicits explicit neuro-symbolic reasoning into 3D MLLM, which benefits the MLLM reasoning capability.

2. The paper conducts extensive experiments to validate the idea, including experiments on different benchmarks and extensive ablation experiments.

3. The method shows strong generalizability by showing the capability of developing new predicates.

Weaknesses:

1. The RL stage (Stage 3) seems to be not connected to the proposed neuro-symbolic framework, additional explanation of why introducing this stage should be added.

2. It is unclear when ablating the RL stage for ScanRefer and Multi3DRefer, whether it is removing the corresponding training dataset, or removing the entire RL stage.

3. Since the method is explicitly trained on ScanRefer and Multi3DRefer training set, how well will it generalize remains unclear, i.e. can it still performs well when trained on ScanRefer but tested on Multi3DRefer? The generalization experiment using only Sr3D and Nr3D seems to be not strong enough since they're from the same set of 3D scenes.

---

> ### Author Rebuttal · Authors · 2026-03-31
>
> ### **W1. Unclear connection between RL stage and neuro-symbolic framework.**
>
> Stage 3 (CoT-RL stage) is central to bridging neuro-symbolic methods
> with 3D MLLMs. Prior NS3D methods depend on dense intermediate
> supervision (every execution step's input/output), available only on
> synthetic datasets. Real-world datasets (ScanRefer, Multi3DRefer) lack
> such annotations entirely. Stage 3 enables the model to *autonomously
> extend* reasoning patterns internalized during Stage 2 (CoT-SFT on
> synthetic programs) to open-vocabulary, complex instructions, guided
> only by outcome-based rewards. It is the mechanism that overcomes the
> closed-set supervision bottleneck of NS3D.
>
> To further analyze where Stage 3 contributes most, we group ScanRefer
> test samples by reasoning complexity (estimated by the number of plan
> steps in the generated CoT) and compare RL vs. SFT-only performances:
>
> | Reasoning Complexity | SFT-only Acc | CoT-RL Acc | Δ |
> |---|---|---|---|
> | Short (≤4 steps) | 47.2 | 45.4 | -1.8 |
> | Medium (5 steps) | 50.7 | 52.2 | +1.5 |
> | Long (≥6 steps) | 45.2 | 47.9 | +2.7 |
>
> A clear trend emerges: **RL benefit increases with reasoning
> complexity**. For simple queries (≤4 steps), SFT from simple synthetic
> programs alone already captures sufficient reasoning patterns, and RL
> exploration introduces marginal noise. For complex queries requiring
> multi-step spatial verification (≥6 steps), CoT-RL provides consistent
> improvement (+2.7%). This might be because CoT-SFT lacks supervised
> traces for longer, more complex programs, while Stage 3's outcome-based
> RL can reward correct longer traces discovered through exploration,
> effectively providing supervision signals for complex reasoning chains
> that CoT-SFT cannot cover. This directly demonstrates Stage 3's role:
> extending systematic neuro-symbolic reasoning to complex, real-world
> instructions where dense symbolic supervision is unavailable.
>
> ### **W2. Ablation setup for RL stage.**
>
> To clarify, when ablating Stage 3, we do not simply remove the data or
> skip the stage. Instead, we add ScanRefer/Multi3DRefer samples to Stage
> 2 as SFT training data with direct answer supervision, rather than
> answer-based CoT-RL. This provides a **controlled comparison** using the
> same data under two paradigms: SFT with standard language modeling
> objectives vs. RL with outcome-based rewards. This isolates the benefit
> of CoT reasoning and the advantage of outcome-based RL for keeping and
> exploring effective reasoning traces when dense intermediate supervision
> is unavailable.
>
> We'll clarify this setup in the revision. Also, we will move the per-stage training data details to the main text in the revision.
>
> ### **W3. Cross-dataset generalization.**
>
> We note that ScanRefer and Multi3DRefer have fundamentally different
> task definitions: ScanRefer guarantees exactly one target object per
> query, while Multi3DRefer allows zero, one, or multiple targets. This
> makes direct cross-evaluation (train on one, test on the other)
> ill-defined without task adaptation.
>
> That said, except the our Sr3D/Nr3D generalization experiments (Table
> 4), we further conduct experiments on SQA3D (spatial QA) and Scan2Cap
> (dense captioning), matching or surpassing prior best methods on both
> (full results in the response to Reviewer arig-W1), The results
> demonstrate the transferability of our proposed framework to different
> 3D scene understanding tasks.
>
>
>
> ### **L1. How strictly the LLM follows neuro-symbolic reasoning patterns?**
>
> We conduct quantitative evaluation on 200 sampled CoT traces (after
> Stage 3 RL) with Claude Opus 4.6 as judge, validated against 50
> human-checked samples (agreement=94%). Please refer to our response to
> Reviewer 2D4M-W2 for the full rubric and table. Key results: 92.5% of
> all traces remain parsable into valid symbolic programs (NS-Compliance),
> and 86.5% maintain consistent object references throughout the reasoning
> chain. Furthermore, we observe that structured reasoning *correlates
> with correctness*: correctly-grounded samples exhibit 88.2% joint
> compliance (both criteria met), compared to 74.8% for incorrect ones
> (+13.4% gap). This indicates that adhering to neuro-symbolic patterns is
> not merely a stylistic property but directly contributes to grounding
> accuracy, and that outcome-based RL in Stage 3 implicitly preserves
> reasoning structure even without explicit symbolic enforcement.

---

> > ### Author Rebuttal · Reviewer_A5U3 · 2026-04-02
> >
> > I appreciate the authors' point-by-point response to my concerns. I believe the additional experiments and clarification makes the work more solid. I'll maintain my positive score.

---

> > > ### Author Response · Authors · 2026-04-07
> > >
> > > Dear Reviewer A5U3,
> > >
> > > We sincerely thank you for your constructive feedback, which has helped us clarify and strengthen our work.
> > > We are encouraged by your recognition of our method's strong generalizability and the extensive experiments.
> > > We are glad that we have fully addressed your concerns and will incorporate discussed clarifications and the new experiment results into the final version.
> > >
> > > Best regards,
> > >
> > > Authors

---

### Official Review · Reviewer_arig · 2026-03-12

**Soundness:** 3
**Presentation:** 3
**Significance:** 3
**Originality:** 2
**Overall Recommendation:** 4
**Confidence:** 3

**Summary:**

This paper studies 3D visual grounding by trying to combine the interpretability of neuro-symbolic methods with the flexibility of 3D MLLMs. The main idea is to distill symbolic programs into natural language chain-of-thought traces, then further improve generalization with RL.

**Compliance With Llm Reviewing Policy:**

Affirmed.

**Key Questions For Authors:**

No further questions.

**Limitations:**

yes

**Strengths And Weaknesses:**

Strength 1. Clear and well-motivated idea.

The paper tackles a real limitation in current 3D grounding methods: symbolic models are interpretable but rigid, while MLLM-based methods are flexible but often unreliable in spatial reasoning. Distilling symbolic execution into CoT is a sensible and interesting way to connect the two.

Strength 2. Performance is good.

The paper compared its method with various previous baselines, and provide comprehensive ablation study. Which verified the effectiveness of the proposed method.

Weakness 1.
The method relies heavily on synthetic symbolic supervision. The Stage 2 training signal appears to be crucial, but it comes from datasets with strong programmatic structure and fully specified intermediate supervision. This raises concerns about scalability: transferring the method to new domains may require substantial manual effort to design symbolic grammars, parsing rules, and data construction pipelines.

Weakness 2.
The reward design is too heuristic and maybe only suitable for this specific scenario, can it be more generalized or simple for general scenarios?

---

> ### Author Rebuttal · Authors · 2026-03-31
>
> ### **W1. Scalability of synthetic symbolic supervision.**
>
> We clarify that our framework is designed to *minimize* the
> reliance on symbolic supervision.
> Stage 2 (CoT-SFT) leverages synthetic programs only to teach general reasoning *patterns* (query decomposition, stepwise spatial verification, etc.), rather than
> domain-specific knowledge. Stage 3 (CoT-RL) then guides the model to apply these
> patterns to open-vocabulary concepts and deeper reasoning structures
> where *symbolic supervision is unavailable*, via outcome-only reinforcement learning.
> This means our framework can in principle adapt to diverse 3D
> understanding task where outcome evaluation is available, less relying substantial manual effort to design symbolic grammars, parsing rules, or data construction pipelines.
>
> To further validate the scalability of our method, we show that our
> paradigm could easily adapt to **different 3D understanding tasks**, by
> simple outcome reward (EM for QA, CIDEr for captioning). We evaluate on
> **SQA3D** \[Ma et al., ICLR 2023\], a situated spatial QA benchmark
> requiring perspective-dependent reasoning and answer generation, and
> **Scan2Cap** \[Chen et al., CVPR 2021\], a dense captioning benchmark
> requiring spatial description of objects in 3D scenes:
>
> | Method | SQA3D EM | ScanRefer CiDEr@0.5 | ScanRefer CiDEr@0.25 | Nr3D CiDEr@0.5 |
> |---|---|---|---|---|
> | LEO | 50.0 | 72.4 | -- | -- |
> | LL3DA | -- | 74.2 | 65.2 | 51.2 |
> | PQ3D | 47.1 | 80.3 | -- | -- |
> | ChatScene | 54.6 | 77.2 | 81.9 | -- |
> | LLaVA-3D | 55.6 | 79.2 | -- | -- |
> | LEGO | 54.8 | 78.6 | 84.7 | 61.4 |
> | Inst3D-LMM | -- | 79.7 | -- | -- |
> | Video-3D LLM | **58.6** | 83.8 | -- | -- |
> | **APEIRIA (Ours)** | **58.6** | **84.1** | **90.6** | **68.1** |
>
> APEIRIA matches or surpasses prior best methods on both tasks despite
> being primarily designed for spatial grounding, demonstrating that our
> curriculum is a general paradigm transferable across tasks without
> modifying the symbolic pipeline.
>
> ### **W2. Reward design is heuristic and scenario-specific.**
>
> While our reward design may appear heuristic, it follows a general paradigm: (1) a format reward (task-agnostic), and (2) an outcome reward that measures task-relevant correctness. The soft grounding reward is one instantiation tailored for spatial grounding tasks sensitive to spatial locations, but the underlying principle, rewarding the model response similarity to the ground-truth output, is broadly meaningful for many 3D tasks. This paradigm naturally extends to other outcome signals, e.g., CIDEr/BERTScore for captioning, or Exact-Match(EM)/LLM-as-Judge for QA. As shown in our cross-task experiments (in the response to W1 above), APEIRIA achieves strong performance on both SQA3D (EM reward) and Scan2Cap (CIDEr reward) by simply adapting the outcome reward, confirming that the paradigm could generalize across diverse 3D understanding tasks without modifying the overall reward design.

---

> > ### Author Rebuttal · Reviewer_arig · 2026-04-05
> >
> > Thank you for providing new results,  I remain my score broadly positive as weak accept.

---

> > > ### Author Response · Authors · 2026-04-07
> > >
> > > Dear Reviewer arig,
> > >
> > > We sincerely appreciate your valuable feedback, which has helped us clarify and strengthen our work.
> > > We are encouraged by that you find our idea clear and well-motivated with good performances.
> > > We are glad that we have addressed your concerns and will incorporate the discussed new task evaluation results, and a more thorough discussion on the scalability to more diverse domains in our final version.
> > >
> > > Best regards,
> > >
> > > Authors

---

### Official Review · Reviewer_2D4M · 2026-03-13

**Soundness:** 3
**Presentation:** 3
**Significance:** 3
**Originality:** 2
**Overall Recommendation:** 4
**Confidence:** 4

**Summary:**

To address issues of transparency in MLLM reasoning and the limited capabilities of the neuro-symbolic framework in an open-vocabulary setting, the authors proposed the APEIRIA framework (neuron-symbolic 3D MLLMs). This framework addresses these two main problems by training the LLM with CoT generated via neuro-symbolic execution, which ensures the LLM's reasoning is transparent and beyond the black box of the baseline model. The training phase includes three stages: Perception Alignment, Symbolic Injection, and Open-Set and Complex Reasoning Generalization. The perception alignment trains the model to comprehend the basics of 3D spatial understanding. The Symbolic Injection is the supervised fine-tuning stage that uses supervision from Chain-of-Thought reasoning generated by a symbolic program.  Lastly, the Open-Set and Complex Reasoning Generalization stage uses GRPO to help the model learn more about spatial grounding via the Soft Grounding Reward function. The APEIRIA framework demonstrates effectiveness on two 3D vision-language benchmarks (ScanRefer and Multi3DRefer) against several neuro-symbolic and 3D MLLM baselines.

**Compliance With Llm Reviewing Policy:**

Affirmed.

**Final Justification:**

Rebuttal addressed my initial concerns regarding the justification of novelty, verification of reasoning trace, and clearer explanation of some unclear parts.

**Key Questions For Authors:**

1. Why do authors limit the evaluation to only two benchmarks? Do the authors plan to include more recent 3D spatial reasoning benchmarks to provide a more in-depth analysis of the proposed framework using the recently developed benchmark?

2. Can the generated response from APEIRIA be used in neuro-symbolic execution, and can the generated neuro-symbolic response be verified for the correctness of the reasoning trace?

**Limitations:**

Yes, the author includes the limitations section.

**Strengths And Weaknesses:**

# Strengths

- The proposed framework illustrates effectiveness across several SOTA models, including Neuro-Symbolic models, Specialist Methods, and even 3D MLLMs.

- The authors provide comprehensive ablation studies to ensure that each proposed component provides significant value to the whole proposed framework.

- The proposed model illustrates the improvement when increasing the parameter sizes, which is promising for extending the method to even larger models.

- The problems that the proposed framework attempted to address are significant and worth exploring, especially the transparency of reasoning.

- The proposed framework is flexible. Some components, i.e., the perception and planning components, can be replaced if a more advanced method is developed in the future, potentially enhancing the effectiveness of the proposed framework.

- The paper is well written and includes some illustrations to help audiences understand the methodology.


---


# Weaknesses

- The evaluation is limited to only two benchmarks. Currently, several 3D Spatial Reasoning datasets have been developed and may provide more in-depth analysis for the framework. For example 3DSRBench[1], OMNISPATIAL[2]. Extending the evaluation to the newly developed benchmarks, which use a more comprehensive task taxonomy, might enhance the paper's effectiveness and provide a more thorough analysis of the model.

- The rewards function of GRPO only focuses on Soft-Grouning Rewards and Format Reward. There is no reward for the neuro-symbolic part, which may be important for ensuring the transparency of the model's generated reasoning trace.

- The pipeline of the APEIRIA framework is unclear during inference time. The example provided in the paper shows a reasoning trace generated purely by the model; however, the author mentions a perception replacement that required access to the generated symbolic instruction. If the model performs all reasoning on its own, this could undermine the claim to have created a model that provides a transparent reasoning trace, which is significant for the paper. Providing an example, even in the appendix of the actual APERIA pipeline, could help resolve this.

- The novelty of the proposed framework seems to be limited. The main novelty of the paper lies in generating Chain-of-Thought reasoning from neuro-symbolic representations. The training stages are widely used, especially for SFT and GRPO.

- The proposed Soft Grounding rewards, while showing the improvement over the IoU rewards, the improvement seems negligible.


[1] Ma, W., et al. (2024). 3DSRBench: A Comprehensive 3D Spatial Reasoning Benchmark.

[2] Jia, M., et al. (2025). OmniSpatial: Towards Comprehensive Spatial Reasoning Benchmark for Vision Language Models.

---

> ### Author Rebuttal · Authors · 2026-03-31
>
> ### **W1,Q1. Limited evaluation benchmarks.**
>
> We thank the reviewer and have carefully examined both benchmarks.
> 3DSRBench and OmniSpatial evaluates spatial reasoning on **2D images or
> video frames** as input, targeting a complementary but distinct problem:
> inferring 3D relationships from projected 2D views (recovering depth,
> handling occlusion and perspective ambiguities). In contrast, our work
> operates on **explicit 3D point clouds** where full geometry is
> available, and the core challenges lie in (1) understanding complex
> spatial relations in 3D scenes, particularly compositional multi-object
> relations; and (2) grounding open-vocabulary natural language to 3D
> geometry, especially in scenes containing dozens to hundreds of objects.
> Evaluating a 3D MLLM on 2D-image benchmarks would conflate these
> distinct capabilities rather than provide meaningful comparison.
>
> We further demonstrate that our paradigm generalizes to **more 3D
> understanding tasks**: we additionally evaluate on **SQA3D** \[Ma et
> al., ICLR 2023\] (situated spatial QA) and **Scan2Cap** \[Chen et al.,
> CVPR 2021\] (dense captioning). APEIRIA matches or surpasses the prior
> best on SQA3D (58.6 EM) and Scan2Cap (84.1/90.6 C@0.5/0.25 on ScanRefer,
> 68.1 C@0.5 on Nr3D). Full table in the response to Reviewer arig-W1.
> This validates that our curriculum distills **task-general reasoning
> patterns** rather than task-specific shortcuts, which is transferable to
> diverse tasks and datasets.
>
> ### **W2,Q2. No neuro-symbolic reward and verifiability of reasoning traces.**
>
> Stage 3 (CoT-RL) does not include a symbolic reward, because dense
> step-by-step supervision is generally unavailable for open-vocabulary
> referring expressions, precisely the bottleneck of prior NS3D methods.
> Our curriculum addresses this: **Stage 2 (CoT-SFT)** internalizes
> reasoning *patterns* from verified symbolic traces; **Stage 3 (CoT-RL)**
> extends them to open-vocabulary instructions via *outcome-based rewards
> only*, without intermediate symbolic annotations.
>
> To verify RL preserves structured reasoning, we evaluate 200 sampled
> traces (after Stage 3) with Claude Opus 4.6 as judge (validated on 50
> human samples, agreement=94%):
>
> | Criterion | Overall | In Correct Samples | In Incorrect Samples | Δ(Correct-Incorrect) |
> |---|---|---|---|---|
> | NS-Compliance: Trace parsable into symbolic program structure | 92.5% | 95.3% | 90.4% | +4.9 |
> | Reference Consistency: objects consistently referenced in the chain | 86.5% | 91.8% | 82.6% | +9.2 |
> | Both criteria met | 80.5% | 88.2% | 74.8% | +13.4 |
>
> Key findings:
> - 92.5% of traces remain parsable into symbolic programs. Though the
>   predicates are now open-vocabulary natural language, going beyond what
>   traditional NS3D frameworks can express or execute
> - correctly-grounded samples show +13.4% higher compliance than
>   incorrect ones, indicating structured reasoning directly contributes
>   to accuracy.
>
> ### **W3. Inference pipeline clarity on modular perception replacement.**
>
> To clarify, the default inference is *fully self-contained*: APEIRIA
> generates the entire trace end-to-end (planning, perception, execution)
> without external modules.
>
> The modular perception replacement is an *optional* enhancement enabled
> by our structured CoT. When it is applied, we inject pre-computed
> `scene()` outputs from a stronger model (e.g., "I see 33 objects: 0 (couch), ..."), after which the model continues
> decoding normally. Subsequent reasoning steps then benefit from more
> accurate perception. This transparent interface enables plug-and-play
> upgrades that prior 3D MLLMs cannot support.
>
> ### **W4. Limited novelty.**
>
> To the best of our knowledge, **APEIRIA is the first to distill
> neuro-symbolic program execution traces into CoT reasoning for 3D
> MLLMs.** This yields unique capabilities unattainable by prior 3D MLLM
> work: (1) **transparent yet flexible reasoning** with each step
> spatially grounded via explicit object references; (2) **modular
> inference enhancement** via plug-and-play component replacement without
> re-training, a capability no existing 3D MLLM possesses; (3) **bridging
> closed-set and open-set vocabularies** through staged distillation
> (CoT-SFT) and exploration (CoT-RL) curriculum, with ablations in Table 3
> confirming each stage is essential.
> Therefore, although SFT and GRPO are established techniques, the novelty lies in the overall framework and
> emerging properties (interpretability + open-vocabulary + modularity).
>
> ### **W5. Marginal improvement of Soft Grounding Reward.**
>
> The desired benefit is more in **training stability and convergence**:
> IoU reward yields 0 for non-overlapping predictions in early
> exploration, leading to sparse reward. Soft grounding provides denser
> gradient signal. Training curves
> ([\[figure\]](https://anonymous.4open.science/r/rebuttal_figure-EFD6/softreward_vs_ioureward_wandb_log.png))
> further confirm soft grounding converges faster, while IoU reward
> eventually narrows the gap but soft grounding maintains a consistent
> lead.

---

> > ### Author Rebuttal · Reviewer_2D4M · 2026-04-04
> >
> > Thanks for detail response to the weaknesses. All of these have resolved my initial concerns. I would increase my score accordingly.

---

> > > ### Author Response · Authors · 2026-04-07
> > >
> > > Dear Reviewer 2D4M,
> > >
> > > We sincerely thank you for your thoughtful feedback throughout the review process, which has helped us clarify and strengthen our work.
> > > And we sincerely appreciate for increasing the score.
> > > We are encouraged by your recognition of our framework's significance, flexibility and comprehensive ablations.
> > > We are glad that we have fully addressed your concerns and will incorporate the discussed new benchmark results, reasoning trace quality analysis, and pipeline illustration in our final version.
> > >
> > > Best regards,
> > >
> > > Authors

---

### Decision · Program_Chairs · 2026-04-30

**Decision:**

Accept (regular)

**Comment:**

The paper introduces the APEIRIA framework, which enhances reasoning in 3D vision-language models by distilling structured neuro-symbolic reasoning.

The main concerns raised include: limited evaluation benchmarks (2D4M); scalability issues of the neuro-symbolic representation (arig, A5U3, 7qpB); limitations in the reward design (2D4M, arig); unclear presentation of certain components (2D4M, A5U3, 7qpB); and questions regarding the level of novelty (2D4M).

The rebuttal addressed most of these concerns, and all reviewers recommended a weak accept. The AC concurs with this assessment.